# Study on Doxorubicin Loading on Differently Functionalized Iron Oxide Nanoparticles: Implications for Controlled Drug-Delivery Application

**DOI:** 10.3390/ijms24054480

**Published:** 2023-02-24

**Authors:** Vladislav R. Khabibullin, Margarita R. Chetyrkina, Sergei I. Obydennyy, Sergey V. Maksimov, Gennady V. Stepanov, Sergei N. Shtykov

**Affiliations:** 1Chemistry Department, Lomonosov Moscow State University, Lenin Hills, 119991 Moscow, Russia; 2State Scientific Center of the Russian Federation, Joint Stock Company “State Order of the Red Banner of Labor Research Institute of Chemistry and Technology of Organoelement Compounds”, 105118 Moscow, Russia; 3Skolkovo Institute of Science and Technology, 143026 Moscow, Russia; 4Dmitry Rogachev National Medical Research Center of Pediatric Hematology, Oncology and Immunology, 117198 Moscow, Russia; 5Center for Theoretical Problems of Physicochemical Pharmacology, 119334 Moscow, Russia; 6Department of Analytical Chemistry and Chemical Ecology, Institute of Chemistry, Saratov State University, 410012 Saratov, Russia

**Keywords:** doxorubicin, sorption, iron oxide nanoparticles, surface functionalization, magnetic properties, cell viability

## Abstract

Nanoplatforms applied for the loading of anticancer drugs is a cutting-edge approach for drug delivery to tumors and reduction of toxic effects on healthy cells. In this study, we describe the synthesis and compare the sorption properties of four types of potential doxorubicin-carriers, in which iron oxide nanoparticles (IONs) are functionalized with cationic (polyethylenimine, PEI), anionic (polystyrenesulfonate, PSS), and nonionic (dextran) polymers, as well as with porous carbon. The IONs are thoroughly characterized by X-ray diffraction, IR spectroscopy, high resolution TEM (HRTEM), SEM, magnetic susceptibility, and the zeta-potential measurements in the pH range of 3–10. The degree of doxorubicin loading at pH 7.4, as well as the degree of desorption at pH 5.0, distinctive to cancerous tumor environment, are measured. Particles modified with PEI were shown to exhibit the highest loading capacity, while the greatest release at pH 5 (up to 30%) occurs from the surface of magnetite decorated with PSS. Such a slow release of the drug would imply a prolonged tumor-inhibiting action on the affected tissue or organ. Assessment of the toxicity (using Neuro2A cell line) for PEI- and PSS-modified IONs showed no negative effect. In conclusion, the preliminary evaluation of the effects of IONs coated with PSS and PEI on the rate of blood clotting was carried out. The results obtained can be taken into account when developing new drug delivery platforms.

## 1. Introduction

Doxorubicin (DOX) and other antibiotics from the anthracycline series attract the attention of the scientific community due to the simultaneous combination of positive and negative effects on human body [1,2,3,4]. Furthermore, DOX plays a key role in the treatment of many neoplastic diseases, in particular cancer, even though its prolonged action causes cardiomyopathy and congestive heart failure [5,6]. A strong desire to reduce cardio- and nephrotoxicity of DOX led to the idea of binding the drug to various nanomaterials, e.g., engineered nanoparticles [7]. Various nano-sized structures based on synthetic and natural polymers, solid lipids and lipid nanoemulsions (liposomes), silver, gold, calcium carbonate, metal–organic frameworks, carbon, silicon, or magnetic materials have been proposed as controlled delivery systems for doxorubicin [8,9,10,11]. Among nanocarbon materials, graphene, graphene oxide, carbon nanotubes, and fullerene are explored [12,13,14,15,16,17], with a focus on the toxicity and biocompatibility of the nanosystems in use. Nanoplatforms based on magnetic nanoparticles are the most commonly used for drug delivery, including DOX [18,19,20,21,22]. Their advantages include simplicity and cheap synthesis, biocompatibility, and the ability to efficiently adsorb the drugs that makes them ideal candidates for the reconstruction of a drug delivery platform. Despite the fact that the different ionic forms produced by iron oxide nanoparticles (IONs) after degradation can be easily assimilated by the body [23], there is still no unambiguous conclusion about the biocompatibility of IONs [24]. This stimulates further research in the field of biological applications of IONs [25,26,27,28,29]. An additional advantage of using IONs is their superparamagnetic properties, which make it feasible to control the drug delivery using an external magnetic field [8,30]. The IONs are also employed in magnetic resonance imaging, molecular imaging, hyperthermia, and cancer diagnostics [19,31,32,33,34,35]. Such attractive biomedical facets open an avenue for combining the specificity of drug delivery and magnetic resonance imaging, which enhances the effectiveness of anticancer therapy [36,37,38].

Due to the developed surface, the IONs tend to aggregate and oxidize. For this reason, their surface is coated with inorganic, organic, or polymeric materials that perform the functions of surface protection and selective functionalization, as reviewed by Zhu et al. [39]. It is important to note that the list of non-polymeric modifiers used for the loading and delivery of DOX is rather short, with a few contributions on IONs coated with silicon dioxide [40,41,42], gold [43], zinc oxide [44], and silicon dioxide activated with glutaric anhydride [45]. In addition, co-coating with carbon and/or gold conjugated with glutathione [46] and polypyrole [47] has been reported. Alternatively, magnetite can be covered with dimercaptosuccinic acid [48], citric acid [49], oleic acid [50], nonionic surfactants [50], and carbon [51].

In contrast, IONs are much more actively functionalized with synthetic and natural polymers for the purpose of DOX delivery [7,52,53,54,55]. Modification of the surface with polymers provides the sorbent with new useful properties, including pH and thermal sensitivity [56,57,58,59] and controlled drug release [60]. The polymer coating based on polyethylene glycol (PEG) helps reduce the drug degradation and thereby reduces the toxic effect on healthy cells [61]. Surface modification with the polysaccharides carrageenan [62], starch [63], cellulose [64], chitosan [65,66], and various conjugates based on polysaccharides [67,68,69,70,71] proved useful to enhance the selectivity of DOX release. The functionalization of IONs by proteins and large biomolecules also contributes to an increased efficiency of the use of IONs in cancer therapy [72,73]. Coating of IONs with heparin promotes anticoagulant activity and also has its own anticancer properties [74,75]. Described in recent literature is the use of nonionic polymers and block copolymers, as detailed in the Appendix A. From the tabulated data, it is evident that the majority of alternative, ionic polymeric coatings contain the carboxyl group that renders a negative charge under sorption conditions and therefore electrostatic interactions with the protonated amino group of DOX. Among other valuable assets, most of the presented polymers are safe for in vivo use and are easily biodegradable.

Another promising direction in shaping cancer therapy is the use of magnetoliposomes [76,77]. Encapsulation of IONs into a liposome reduces the toxicity of the DOX and improves the IONs delivery, increases the DOX release, and also expands the possibilities for combining treatment methods [76,78,79,80]. However, there is a need to destroy the liposome prior to applying the combined chemotherapy with magnetic hyperthermia, which requires additional technical equipment [81].

It should be noted that sometimes the modifier adversely affects the biocompatibility of the material. The use of a cationic surfactant, CTAB, as a surface modifier increases the loading of DOX but reduces cell viability, causing the formation of pores in the cell membrane [82]. Therefore, the major aim of coating is to balance high sorption capacity, selectivity, and aggregative stability of nanoplatforms with their non-toxicity.

To reach the affected tissue, the IONs must pass through the capillaries and avoid phagocytosis, thus, their size should be less than 100 nm [83,84]. In addition, IONs must have a high magnetization which should disappear after the removal of the magnetic field in order not to cause the formation of aggregates [83,85]. Accordingly, the average size of IONs less than 100 nm or in the range of 100–150 nm is considered the best for loading and efficient drug delivery [7,86]. When choosing the proper size of IONs, it is also necessary to take into account the methods of their excretion from the body. It has been reported that particles up to 20 nm in diameter are removed from the blood through the kidneys, while particles > 100 nm, as a rule, are absorbed by the liver, spleen, or bone marrow [84].

A detailed analysis of available literature allows for several general conclusions. First, commonly used are the IONs of widely varied sizes, from small 6–10 nm, to medium (tens of nm), and quite large, over 100 nm and up to 300 nm. This is due to the fact that a priori it is unclear which particular size would provide the best targeting function [84,87,88]. For instance, for IONs ranging from 60 to 310 nm it was found that smaller nanoparticles demonstrate better cellular internalization, deeper penetration into multicellular spheroids, and provide higher efficiency of photothermal ablation in vitro. At the same time, large IONs are better accumulated in tumors, causing more effective inhibition of their growth. In their turn, 120 nm particles are best suited for magnetic resonance imaging and in vitro photoacoustic tomography. It is noted that apart from the size, the particle shape and the chemical composition of the modified surface may be important [89].

Second, in most studies the authors focus on block copolymers of complex structure (see Appendix A). However, such techniques are of little use in the biomedical practice, since multi-stage synthesis is difficult and time-consuming. Furthermore, sorption and desorption processes on such modified particles sometimes last several days and have low efficiency. Third, only one type of ION surface modifier is usually subject to investigation. This shortcoming retards the choice of the most effective carrier for DOX. Likewise, little attention has been paid on the influence of the size and shape of the ION core.

The aim of this work was to (i) obtain IONs of 80–150 nm in size, (ii) modify their surface with cationic and anionic polyelectrolytes, nonionic polymer dextran, and porous carbon, and (iii) reveal the optimal core–shell system for the sorption of DOX, with the objective to develop and implement an effective nanocarrier for controlled drug delivery. The modified IONs were characterized in detail by various methods (XRD, FTIR, measuring the zeta-potential, scanning and transmission microscopy, etc.) to explain the sorption behavior.

## 2. Results and Discussion

### 2.1. Characterization of IONs

#### 2.1.1. X-ray Diffraction

The crystal structures of Fe_3_O_4_ were characterized by X-ray diffractometer, as shown in Appendix A. Diffraction peaks at 2θ values around 30.11° (200), 35.61° (311), 43.13° (400), 53.67° (422), 57.17° (511), 62.77° (440), and 73.25° (533) correspond to the lattice planes of the face-centered cubic Fe_3_O_4_ phase [90,91], being in good agreement with the standard XRD data card of Fe_3_O_4_ crystals (JCPDS No.85–1436). The synthesis product, in addition to magnetite, contains fractions of hematite Fe_2_O_3_ and goethite FeO(OH), at 24 and 6%, respectively. High-temperature treatment (boiling) in propylene glycol reduced the proportion of hematite to 11%, while the goethite phase disappeared completely.

#### 2.1.2. SEM and TEM

The results of TEM and SEM measurements of particle size are presented here and in the Appendix A. The synthesized materials are a polydisperse ensemble of crystalline particles of cubic magnetite. Particle size (*d*) for unmodified magnetite Fe_3_O_4_, determined from TEM and SEM images, is in the range of 80–150 nm. The calculated values of *d* were limited to a sample of 500 particles. The presence of fine (*d* < 80) and larger (*d* > 150) fractions does not exceed 10% by volume. Thus, the average particle size (*d*_av_) is 115 ± 35 nm. The TEM micrographs also show variable in size aggregates with a size of 300–400 nm.

Using HRTEM, the size of the magnetite particles after surface modification (PSS, PEI, and Carb) was determined to estimate the change in the size of the shell thickness after the sorption of DOX (see Section 3.6 for more detail). Based on TEM images, it was found that the shell for PSS- and PEI-modified IONs is uniform (Figure 1c,e). The thickness of MNPs is 11.6 ± 0.6 and 11.2 ± 2.6 nm for Fe_3_O_4_@PSS and Fe_3_O_4_@PEI, respectively. The thickness of the porous carbon layer in Fe_3_O_4_@Carb (Figure 1g,h) is substantially wider, 26.0 ± 6.7 nm. The TEM micrographs witness that the IONs coated with PSS and PEI are in an aggregated state while forming the chain structures (Appendix A). The reason may be the interaction between the polymer shells of MNPs as well as intrinsic limitations of TEM and SEM, operating in a high vacuum where particles tend to stick together.

For unmodified magnetite, the particle sizes before (Figure 1a) and after DOX sorption (Figure 1b) are comparable. The presence of the drug was not detected on HRTEM-photographs, and the change in particle size is not significant, amounting less than 5% of the average size. The results for the modified materials also indicate the absence of noticeable changes in the particle morphology after the sorption of DOX: the shell thickness of Fe_3_O_4_@PSS@DOX was 11.0 ± 0.7 nm (Figure 1d), and for Fe_3_O_4_@Carb@DOX 26.1 ± 7.5 nm (Figure 1i). Changes for Fe_3_O_4_@PEI are more noticeable. After the sorption of the drug, the average shell thickness increased by 2.9 nm (the thickness of the Fe_3_O_4_@PEI@DOX shell was 14.1 ± 1.9, see Figure 1f), likely indicating the presence of DOX on the surface.

#### 2.1.3. Infrared Spectroscopy

The IR spectra of the original magnetite and magnetite coated with porous carbon and polymers are shown in Figure 2. The summary of peak assignment, given in Table 1, confirms the successful surface modification.

The IR spectra of DOX itself and magnetite nanoparticles with adsorbed DOX are presented in Figure 3. In all cases the drug’s spectrum is subject to drastic changes, indicating that its sorption does take place. The strongest spectral changes are observed upon sorption on the pure magnetite nanoparticles and Fe_3_O_4_ coated with porous carbon. Characteristic absorption peaks and their assignments are summarized in Table 2.

The change in the IR spectra after the sorption of DOX indicates the presence of the drug on the surface of the sorbents.

#### 2.1.4. Magnetic Characterization

The magnetic properties were measured using vibrating-sample magnetometry that assumes that the saturation magnetization (*Ms*) is related to the sample mass [109]. By a slightest change in the *Ms* and coercive force (*Hc*), it is possible to detect not only changes in the internal structure of the magnetic material and pertinent chemical processes (particle growth, oxidation, etc.), but also changes in the mass of the sample with high accuracy [70,110].

As can be seen from Appendix A and Table 3, implementation of the modifiers slightly reduces the magnetization compared to the parent magnetite. Specifically, functionalization with polymers, regardless of their nature, results in a magnetization decrease of almost 10%. This may indicate small differences in the thickness of the modifier layer on the magnetite surface, shielding the magnetic core and hence reducing its magnetic moment [69]. For Fe_3_O_4_@Carb, the magnetization is reduced by 20%. Importantly, for all types of nanoparticles under scrutiny, superparamagnetic behavior is observed with a characteristic absence of residual magnetization and an insignificant *H_C_* of 5.0 mT. This phenomenon presents a crucial factor for using magnetic drug carriers in the biomedical field. As already mentioned above, one of IONs’ positive features is that after the removal of the external magnetic field, there is no residual magnetization, which causes the aggregation of magnetic particles. After the sorption of DOX, a slight change in the saturation magnetization was observed (Table 3) and additionally to IR-spectra and HRTEM images, this finding indicates the presence of the drug on the adsorbent and an increase in the thickness of the nonmagnetic layer of the particle. The change in *Ms* was no more than 4%. The greatest change is observed in PEI- and carbon-modified IONs.

#### 2.1.5. Zeta-Potential Measurements

Results of determining the zeta-potential of functionalized IONs are presented in Figure 4. In all cases, surface modification significantly changes the *ζ* of the initial magnetite. For the polyelectrolyte-modified particles, the *ζ* exceeds the accepted stability threshold (𝜁 > |30| mV [111]) at most of the examined pH values so that one could expect their improved stability behavior in suspensions. Nanoparticles coated with porous carbon have a negative ζ over the entire pH range, but only at pH ≥ 10 does its value (−34 mV) correspond to the mentioned stability condition. The most plausible reason of this observation is the presence (and dissociation in an alkaline medium) of carboxyl and hydroxyl groups on the surface of porous carbon. Coating with dextran reduces the surface charge by virtue of the particle enveloping with a polymer shell, built-up via the mechanism of hydrogen bond formation [112]. At the same time, in the region of pH 5.5, an isoelectric point was observed.

### 2.2. Assessment of Colloidal Stability

An important characteristic for dispersion of nanoparticles is stability over time. The stability of magnetite suspensions might be low because of rather large particle sizes. Here, to assess the sedimentation stability, the method of sedimentation in a gravitational field presented in the Appendix A (Procedures) was used [113]. To evaluate the effect of modification chemistry on colloidal stability, the time during which the particle sedimentation takes place was measured (Appendix A). The data obtained, as well as the results of measuring the hydrodynamic diameter *d*_av_ (Z-average size) of samples, are collected in Table 4.

The DLS measurements give only an approximate size of the MNPs because they are very sensitive to temperature, solvent viscosity, and the presence of “shell” around particles [114]. For TEM, the dried samples are to be used, since the method works under ultra-high vacuum conditions. For this reason, the average particle size obtained by DLS is usually higher [114]. Diagrams of the distribution of particle size by intensity are presented in Appendix A.

The data of Table 4 show that in all cases the stability of MNPs is not high and it is limited to 11 min. On the other hand, the tested nanomaterials demonstrate a different sedimentation rate, which indicates the important role of the surface modifier. The sedimentation time, zeta-potential, and hydrodynamic size of Fe_3_O_4_@PEI remain practically unchanged at both pH values and their sedimentation proceeds quite slowly due to the high 𝜁. The Fe_3_O_4_@PSS particles have the best stability in suspension at pH 7.4 due to the complete dissociation of modifier’s sulfonic groups. Unusual increased stability in the colloidal solution of IONs coated with dextran at pH 7.4, despite the lowest 𝜁 values, can be explained by the steric factor [115]. On the other hand, the Fe_3_O_4_@Carb suspension displays marginal stability (see Table 4), as an effect of large particle size and low zeta-potential. Changes in the rate of colloidal stability after sorption of DOX were not observed. The value of the sedimentation rate of IONs loaded with DOX did not change significantly and was within experimental error (see Table 4, values in parentheses).

Thus, based on the results of Section 2.1.3 and Section 2.1.5 on the study of the sorption of various modifiers on the surface of MNPs, as well as the literature data on the state of bare magnetite nanoparticles and polymers in an aqueous solution at pH 7.4, we can assume that this sorption is caused by the complex various types of interactions. Since, at this pH, the amino groups of PEI are partially protonated, and the OH groups on the magnetite surface are partially dissociated, its binding to the polyelectrolyte is due to both electrostatic interaction and the participation of the H-bond [91,116]. In the case of PSS, the negatively charged sulfonic groups of the polymer interact electrostatically with surface iron cations and simultaneously form H-bonds with the surface OH groups of the magnetite. The binding of dextran and porous carbon to magnetite is also caused by electrostatic forces, van der Waals interactions, and H-bonding [99,100,101,102,117].

### 2.3. Sorption of Doxorubicin

Figure 5 shows typical sorption isotherms and dependences of the degree of sorption on the mass of the sorbent. The highest degree of sorption (65%) is achieved when using the Fe_3_O_4_@PEI nanoparticles, the outcome being in accord with their highest zeta-potential (+53 mV). It can be assumed that under these conditions, sorption takes place through the formation of hydrogen bonds between non-protonated amino groups of PEI and heteroatoms of the phenolic, alcohol, methoxy, and amino groups of DOX. According to Suh et al. [118], at pH 7.4 about 80% of amino groups of PEI, which can participate in the formation of H-bonds with DOX, are not protonated. In turn, at this pH about half of the DOX amino groups are also not protonated [119,120,121]. Our assumption is consistent with the findings of a study by Coluccini et al. [122], who by applying the ^1^H NMR spectroscopy as well as NOESY and NOE NMR analysis in D_2_O found out that the chemical shifts of the protons of the proton-donor groups and aromatic rings of DOX, PEI are all subject to alterations upon the addition of PEI.

A higher degree of drug loading onto Fe_3_O_4_@Carb (compared to unmodified IONs or Fe_3_O_4_@Dex) is presumably associated with the shell porous structure. Increased negative potential favors the electrostatic interaction of IONs with the drug cation formed due to the partial protonation of the DOX amino group [121]. Such interpretation is consistent with an even a higher degree of sorption (32%) by Fe_3_O_4_@PSS particles, whose negative zeta-potential at pH 7.4 is higher than that at pH 5.0. An additional factor for increasing the degree of sorption by Fe_3_O_4_@PSS may be the hydrophobic interaction of modifier’s benzene rings with the hydrophobic parts of the drug molecule. From the data of Figure 5b, it follows that a 95% sorption of DOX (0.5 mg) into nanoparticulate form requires from 2 mg Fe_3_O_4_@PEI and 15 mg Fe_3_O_4_@PSS to 25 mg Fe_3_O_4_@Carb and more than 30 mg Fe_3_O_4_@Dex (all sorbents being considerably more efficient than Fe_3_O_4_).

To gain a deeper understanding of the characteristics of the synthesized sorbents, we compared them with other reported nanosized sorbents used for loading DOX (see Appendix A). The PEI- and PSS-functionalized IONs of interest have a higher sorption capacity (691 and 325 mg·g^–1^, respectively). Such a difference can be explained by the fact that in the case of PEI, the interaction with DOX involves all oxygen-containing groups of DOX and amino groups of PEI, whereas PSS can interact electrostatically only with the protonated amino group of DOX. This explains the larger amount of the PSS-sorbent required for the sorption of DOX.

Another benefit of the modified IONs is much shorter time required for making quantitative drug loading true. However, the most principal advantage comprises the potential of magnetic site-specific targeting, when the drug is delivered to the deceased tissue or organ by means of external magnetic field.

### 2.4. Adsorption Kinetics

Comparison of the efficiency of DOX sorption on all sorbents of our interest follows from Figure 6, with all the fitting kinetic parameters listed in Table 5. The initial adsorption stage is rapid as the DOX molecules incline to be bound onto an external, highly developed surface of the nanoparticles. The correlation coefficients (*R*^2^) presented in Table 5 witness that the pseudo-second-order kinetic model describes the process of DOX adsorption more precisely. This confirms that adsorption depends on the amount of the drug and the active sites on the surface of the sorbent. It is interesting to note that the calculated values of *q_e_*, determined by the pseudo-second-order model, are larger than the experimental ones (Table 5). This may indicate the need for employing longer sorption times.

Thus, immobilization of DOX on nanoparticles may take place due to the electrostatic and van der Waals interactions, hydrogen bonds, and π–π stacking of the anthracycline DOX fragment with aromatic fragments of the coating material [122]. Thus, it is difficult to predict what factor(s) governs the efficiency and degree of immobilization.

### 2.5. Desorption of DOX

An important step in developing the ION-based drug delivery system was the verification of drug release. To study the applicability of modified IONs as carriers of anticancer drugs, the behavior of DOX release was modeled at physiological (pH 7.4) and cancer cell pH (pH 5.0) The resulting kinetic profiles are depicted in Figure 7 and Appendix A, the latter portrays the first 60 min of desorption.

As noted in many studies, desorption of DOX at pH 7.4 proceeds 2–4 times slower than in a slightly acidic environment [10,123,124,125]. The same behavior was observed in our experiments. As shown in Figure 7a, all samples showed no significant drug release (<7%) at pH 7.4. This collective behavior of sorbents is associated with a strong electrostatic interaction between sorbent and sorbate. Additionally, this may be due to the poor solubility of the drug itself with an increasing the pH [126]. At the same time, abrupt initial desorption (Appendix A), which is observed in all samples during the first 15–30 min, is associated with the weak physical adsorption of some drug molecules on the surface of MNPs.

On the other hand, as the pH decreases, as shown in Figure 7b, drug desorption increases. In all cases, at pH 5.0, the drug is only partially desorbed, with a highest release of about 30% recorded for the PSS-modified IONs. It is important to mention that within the first 30 min, i.e., the timeframe simulating in vivo application of a magnetic nanoformulation, DOX release reaches up to 15%. This would generate drug active concentration of 0.14 mmol·L^−1^ and 2–3 times higher that could be achieved with other nanosorbents [127,128,129].

The IONs coated with Dex and Carb demonstrate relatively easy desorption in an acidic environment (but not to the full extent). In the case of Dex, two phenomena can play a role here. On the one hand, the zeta-potential (in absolute units) decreases with decreasing the pH and passes through the isobestic point at the region of pH 5.5, hence implying a weakening of electrostatic interactions and increasing the release of the drug. Similar results were observed by Liu et al. [69] for the DOX desorption from the surface of modified carboxymethylchitosan. On the other hand, a decrease in pH reduces the solubility of Dex-modified IONs (see Section 2.2) and increases their aggregation [115], which can hinder the drug release over time.

In the case of the Carb coating, we assume the presence of two types of localization of the drug on the surface. In the first case, DOX is attached to the surface due to electrostatic forces, and a change in the zeta-potential led to a decrease in forces and desorption of the drug. In the second case, the drug enters the cavities of porous carbon layer, making desorption extremely difficult to achieve. It is possible that additional conditions are required for such desorption mechanism such as the magnetic-field-induced drug release [130].

In our opinion, a slow (but sufficient) release of the drug can be considered as a positive factor, ensuring its long and uniform cell-killing action. In therapeutic practice, this would allow for preventing repeated exposure, drug overdosage, and possibly reducing its negative side effects. From this viewpoint, the Fe_3_O_4_@PEI nanoparticles may have the greatest medicinal potential, with low release efficiency compensated by high sorption capacity.

### 2.6. Cell Viability Analysis

The cytotoxicity of DOX and IONs is well known. Therefore, our relevant interest was directed to the modifiers PSS and PEI. Whereas PSS in the free form is well studied, for instance in the treatment of hyperkalemia [131], where it showed no measurable cytotoxicity [132,133], PEI proved to be a highly toxic agent capable of initiating apoptosis and cell necrosis [134,135,136]. Moreover, as noted by Kafil et al., branched PEI is more harmful than a linear analog [134]. Furthermore, it was shown that an increase in the molecular weight of PEI inhibits the cell proliferation [137].

The widely used MTT assay has a number of limitations and is only capable of assessing cytotoxicity based on the metabolic activity of the cell according to the “living–non-living” principle [138]. At the same time, the mechanism by which this or that agent affects the cells remains beyond the scope of the analysis. It is also impossible to transfer the properties of individual components to the entire system, which forced us to consider modifiers not as separate substances, but in combination with the core entity. The results acquired revealed that the presence of the Fe_3_O_4_@PEI and Fe_3_O_4_@PSS NPs in the cell medium (for both materials) did not statistically reduce cell survival. Figure 8 shows the percentage of living cell depending on the concentration of particles added to the cell medium. It is clear that all of the tested particles and concentrations did not affect cells viability remaining no less than 75–80%.

The most intriguing result is the impact of small doses of IONs (5 µg·L^−1^) and PEI-modified IONs (5 and 1 µg·L^−1^), as statistical analysis by non-parametric one-way ANOVA indicated their difference with the control group. Probably, this could be related to the better availability of particles for cells at small concentrations compared to applying higher concentrations, at which the particles can agglomerate and become unavailable to cells.

### 2.7. Real-Time Platelet Dynamics

Biocompatibility testing involves a systematic approach with many different assays. As a preliminary trial, we studied the effect of IONs on the rate of thrombus formation. Fe_3_O_4_ nanoparticles, as previously found, have high biocompatibility with blood and are widely used in vitro and in vivo assays [25,139]. The presence of magnetite nanoparticles does not cause degradation of blood cells (leukocytes, thrombocytes, and erythrocytes) or noticeable side effects [140]. However, in our case, MNPs have a developed surface, imposing various interactions with blood cells. The rate of thrombus formation on the surface of glass coated with collagen was assessed during the first 10 min for three different MNPs: Fe_3_O_4_@PSS, Fe_3_O_4_@PEI, and Fe_3_O_4_. Blood without the addition of nanoparticles was used as a control.

For the control sample, the rate of thrombus formation, as well as the size of the formed thrombi, is comparable to each other (Figure 9). At the same time, the blood flow through the capillary was laminar and the speed constant. Abnormal aggregative (or other) activity was not observed. Similar results were obtained for unmodified and PSS-modified nanoparticles (see Figure 9). By the end of the analysis, the thrombi reached comparable sizes, similar to the results of the control experiment.

Conversely, Fe_3_O_4_@PEI caused increased thrombus formation. The appearance of large aggregates of platelets and immune cells was already registered in the blood at the entrance to the flow chamber with collagen, which may indicate the effect of this sample type of particles on platelets. The morphology of the formed thrombi was visually different from other samples, and their size was much larger than in other cases.

## 3. Materials and Methods

### 3.1. Chemicals

Iron(II) sulfate, ammonium chloride, 25% ammonia solution, and o-phosphoric acid were purchased from Reakhim (Moscow, Russia). Sodium hydroxide was obtained from LenReaktiv (St. Petersburg, Russia). 1,2-Propylene glycol was the product of EKOS-1 (Moscow, Russia). D-Glucose monohydrate was obtained from LenReaktiv (St. Petersburg, Russia). Anhydrous sodium hydrogen phosphate (≥99.0%) and potassium dihydrogen phosphate (≥99.5%) used to prepare the buffer solutions were purchased from Sigma-Aldrich (St. Louis, MO, USA). All chemicals were of analytical grade and were used as received. Dextran ((C_6_H_10_O_5_)_n_ M_r_~40.000), sodium poly(4-styrenesulfonate) (PSS, M_r_~70.000) and branched polyethylenimine (PEI, M_r_~25,000) were obtained from Sigma-Aldrich, DOX (lyophilizate for solution for injection, 10 mg/ampoule) was obtained from Pharmachemie (Petah Tikva, Israel). All solutions were prepared using ultrapure water obtained from a Milli-Q system, Millipore Corporation (Millipore SAS, Molsheim, France).

### 3.2. Synthesis of IONs

Cubic magnetite (Fe_3_O_4_) was obtained by precipitation followed by deposit aging in accordance with slightly modified published approach [141] (see Figure 10 for details). First, an alkaline solution was prepared by dissolving 3.86 g of NaOH in 400 mL of deionized water under nitrogen atmosphere. Then, a solution of iron(II) sulfate was made up by dissolving 10.0 g of FeSO_4_·7H_2_O in 100 mL of deionized water under nitrogen atmosphere. This solution was added to the NaOH solution heated to 80 °C and constantly stirred at 2000 rpm. Next, a current of air (1000 mL·min^−1^) was passed through the mixed solution for 2 h, as a result the colloidal solution changed color from light blue to black. The resulting IONs were isolated from suspension by centrifugation (6000 rpm, 10 min), washed several times with deionized water and ethanol, and dispersed in propylene glycol under the action of ultrasound for 1 h. After heating at 180 °C for 3 h in a Teflon autoclave, the particles were cooled, washed with water and ethanol, and isolated by centrifugation (same conditions as above). The IONs were applied at a concentration of 100 μg·mL^−1^ in all suspensions under investigation, unless stated otherwise.

### 3.3. Coating with Polyelectrolytes and Dextran

An aqueous suspension of IONs was mixed with a respective modifier solution (100 μg·mL^−1^) at a ratio of 1:10, sonicated for 10 min, and stirred for 24 h at 50 rpm. The modified particles were washed several times with water and ethanol and isolated by centrifugation at 4000 rpm (Figure 11).

### 3.4. Coating with Porous Carbon

Nanoparticles were coated with porous carbon (Carb) by hydrothermal carbonization in aqueous medium [117]. A total of 10 mL of an aqueous particle suspension (10 mg·mL^−1^) was added to 20 mL of water containing 200 mg of glucose, at a ratio to saccharide as 1:2 by weight. The mixture was diluted with deionized water to 50 mL, stirred for 2 h and then sonicated for 20 min. The resulting suspension was quantitatively transferred to a Teflon autoclave, kept for 12 h at 180 °C, cooled, and washed several times with water and ethanol. Finally, the modified particles were separated by centrifugation at 4000 rpm (Figure 12).

### 3.5. Characterization of Nanoparticles

The size and shape of nanoparticles were measured using a scanning electron microscope with a resolution of 1.2 nm at 30 keV, the VEGA 3 Tescan (Brno, Czech Republic) and the transmission electron microscope Tecnai-G2 F20 (FEI Company, Hillsboro, OR, USA) with a spatial resolution of up to 0.2 nm. Hydrodynamic diameter (z-average size) and surface charge (zeta-potential, *ζ*) values of sample suspensions were recorded by dynamic light scattering (DLS) analysis using a Nano-ZS Zetasizer, model ZEN3600 (Malvern Instruments Ltd., Malvern, UK) at an angle of 173°. X-ray diffraction measurements were carried out on a Bruker D2 PHASER diffractometer (Bruker, Billerica, MA, USA) using Cu Kα radiation (*λ* = 0.154 nm) at 40 kV and 30 mA in the range of 2θ values (from 10° to 80°). Absorption and IR spectra were recorded with a Shimadzu UV-2550 spectrophotometer (Shimadzu, Kyoto, Japan) and a Bruker Vector 22 FTIR spectrometer (Bruker, Ettlingen, Germany), respectively. Magnetization curves were monitored using a Lake Shore 7407 magnetometer (Lake Shore Cryotronics Inc., Westville, IN, USA). The solutions were mixed on Bio RS-24 analog controlled rotator (BioSan, Riga, Latvia). Used for the magnetic separation was a permanent Nd–Fe–B magnet with (VN)_max_ = 40 MGOe (Guangzhou, China). For dispersion and functionalization of IONs, a Grad 57–35 serial ultrasonic bath with a generator power of 165 W (Grad-Technology, Moscow, Russia) was used. For hydrothermal carbonization, a 50 mL Teflon autoclave with a Toption (Xi’an, China) stainless steel body was employed. Sedimentation analysis and sample centrifugation were performed, respectively, on an Vibra HT-224RCE analytical balance Shinko Denshi, Co. (Tokyo, Japan) and using an EBA 200 centrifuge (Hettich, Kirchlengern, Germany).

### 3.6. Sorption of DOX

An analysis of the literature showed that the optimum conditions for the sorption of DOX correspond to neutral phosphate-buffered saline (PBS; Appendix A). Polypropylene tubes were used for sorption, as previously recommended [42]. To assess the effect of sorption time, 5 mL of standard ION suspension and 5 mL of aqueous solution of doxorubicin (100 μg·mL^−1^) were placed in each of six test tubes and the mixture was diluted to 30 mL with phosphate buffer solution at pH 7.4 (10 mmol·L^−1^). The tubes were placed in a rotary mixer and stirred at 20 rpm. At certain time intervals (2, 5, 10, 15, 30, and 45 min), one test tube was removed and after the precipitation of IONs by applying an external magnetic field, the supernatant was separated to determine the unbound drug spectrophotometrically [142].

To assess the effect of the nanosorbent mass, 8 solutions were used, containing the same amount of doxorubicin (500 μg) and increasing amounts of IONs, from 0.5 to 50 mg (final volume 10 mL). The tubes were placed in a rotary mixer and stirred for 45 min at 20 rpm. The IONs were then magnetically separated and the concentration of DOX in the supernatant determined (Figure 13).

The degree of sorption was calculated as
(1)$$R=\frac{\left(C_{0}-C\right)}{C_{0}}\cdot 100\%$$
where *C*_0_ is the initial concentration of the drug and *C* is its residual concentration after sorption. To compare the efficiency of drug sorption by different sorbents, we used the sorption capacity calculated by the following equation:(2)$$q=\frac{{\left(m_{0}-m\right)}_{DOX}}{m_{S}}$$
where *q* is the difference between the initial (*m*_0_) and residual (*m*) masses of DOX in solution and *m_S_* is the mass of the sorbent. Each experimental series was repeated at least five times.

We repeated the measurements three times and built graphs of the averaged values.

### 3.7. Adsorption Kinetics

To evaluate the adsorption kinetics, we used the pseudo-first-order and pseudo-second-order models [109] specified by the following equations:(3)$$q_{t}=q_{e}\left[1-exp(-k_{1}t)\right]$$
and
(4)$$q_{t}=\frac{q_{e}^{2}k_{2}t}{1+q_{e}k_{2}t}$$
where *q_t_* is the amount of adsorbed drug at time *t* per 1 g of sorbent (mg·g^–1^), *q_e_* is the equilibrium (maximum) amount of adsorbed drug per 1 g of sorbent (mg·g^–1^), *k*_1_ (min^–1^), and *k*_2_ (g·mg^–1^·min^–1^) are the pseudo-first and pseudo-second order constants, respectively.

### 3.8. Desorption of Doxorubicin

In the release studies, 5.0 mg of differently functionalized IONs with adsorbed drug were placed in 50 mL of phosphate buffer (pH 5.0 and 7.4) and mechanically stirred for 24 h at 20 rpm. At the pre-determined time points (5, 10, 15, 30, 60, 120, 360, and 1440 min), a 5 mL aliquot was removed, the supernatant was separated from the sorbent using a magnet and the doxorubicin concentration was determined by the spectrophotometric method. DOX releases were calculated using the following equation:(5)$$Cumulative\;release=\frac{m_{t}}{m_{\infty }}\cdot 100\%$$
where $m_{t}$ is the amount of released DOX at time *t* and $m_{\infty }$ is the total amount of DOX loaded onto the sorbent.

Each experimental series was repeated at least three times and the averaged values were used to build graphs.

### 3.9. Cell Viability Analysis

Cell viability analysis was carried out according to the method described earlier [143], with slight changes.

#### 3.9.1. Cell Culture

Neuro-2A (N2A) cell lines (obtained from the American Type Culture Collection, ATCC, Manassas, VI, USA) were cultured in standard DMEM media (Biolot, St. Petersburg, Russia) supplemented with 10% fetal bovine serum (FBS; Gibco, Waltham, MA, USA), L-glutamine (Biolot), and 100 μg·mL^−1^ penicillin/streptomycin (Biolot). The media were replaced every 3 days, and the cells were maintained in a humidified incubator (Innova CO-170, Hyland Scientific, Washington, DC, USA) at 5% CO_2_ and 37 °C.

#### 3.9.2. Cell Viability

The effects of the nanoparticles on N2A cells were determined by the standard AlamarBlue test. Cells were seeded into 96-well cell culture plates (Eppendorf, Hamburg, Germany) at a cell density of 10^4^/well in the culture medium and incubated at 37 °C under 5% CO_2_ during 24 h. The particle solutions at concentrations 1, 5, 10, 50, and 100 µg·L^−1^ were tested. After 24 h, 10 μL of the fluorescent dye AlamarBlue (10,000 U·mL^−1^, Thermo Fisher Scientific, Waltham, MA, USA) was added to each well, and the fluorescence (540/590 nm) intensity was measured with a spectrophotometer (Infinite F200 PRO, Männedorf, Switzerland).

### 3.10. Real-Time Platelet Dynamics Ex Vivo Observed by Confocal Microscopy

The rate of thrombus formation in the presence of NPs was assessed according to a slightly modified method described in [144].

#### 3.10.1. Human Blood Collection

Human blood collection investigations were performed in accordance with the Declaration of Helsinki under a protocol approved by the NMRC PHOI Ethical Committee Informed written consent was obtained from all donors. Blood was collected in sterile tubes containing Hirudin anticoagulant (100 U·mL^−1^; Merck, Darmstadt, Germany).

#### 3.10.2. Preparation of Flow Chamber System and Human Blood for Perfusion

Glass coverslips (24 × 24 mm; Heinz Herenz Medizinalbedarf GmbH, Hamburg, Germany) were cleaned with Plasma Cleaner (Harrick Plasma Inc, Ithaca, NY, USA). In order to immobilize collagen fibers on the surface of cleaned glass coverslips, we diluted stock solution of native Chronolog fibrillar type I collagen reagent (1 mg·mL^−1^, Chrono-Log, Havertown, PA, USA), which already has collagen fibrils, with a 20 mM acetic acid (Reakhim) to a final concentration of collagen solution of 200 µg·mL^−1^. This solution was added as a 10 µL drop to the surface of a cleaned glass coverslip and incubated for 40 min in a humid chamber at room temperature, then rinsed with water, dried, and assembled as part of the parallel platelet flow chamber described in [145]. Before perfusion, the fluorescent dye DiOC_6_(3) (3,3′-dihexyloxacarbocyanine iodide; Thermo Fisher Scientific) was added in an amount of 0.1% (final concentration 1 µM) of the blood volume and an aqueous dispersion of IONs (concentration 0.5 µg·µL^−1^) in an amount of 1% of the blood volume.

#### 3.10.3. In Vitro Flow-Based Thrombus Formation Assay

Hirudinized (100 U·mL^−1^) human whole blood in the presence of DiOC_6_(3) dye was perfused at a wall shear rate (1000 s^−1^) through collagen-coated channels at room temperature using a programmable syringe pump PHD 2000 (Harvard Apparatus, Hollistion, MA, USA). DiOC_6_(3)-loaded platelets were visualized by differential interferential contrast (DIC) or fluorescence microscopy with an Axio Observer Z1 microscope (Carl Zeiss, Jena, Germany) equipped with a 100× microscopic objective. Images were acquired with a Photometrics EMCCD camera (QuantEM 512sc, Teledyne Technologies, Thousand Oaks, CA, USA).

## 4. Conclusions

Thus, in this work, we functionalized the nanomagnetite surface with cationic (PEI), anionic (PSS), and nonionic (dextran, Dex) modifiers, as well as porous carbon (Carb), and compared the degree of loading of these and bare MNPs with doxorubicin. Using DLS and TEM, we found that the size of the obtained sorbents is in the range of 80–290 nm. We have shown that modification with polymers increases the stability of particles in aqueous suspension by a factor of 4–6 compared to unmodified magnetite. A comparison of the sorption capacity of MNPs with respect to DOX made it possible to establish the following series of sorption efficiency: Fe_3_O_4_@PEI (691) > Fe_3_O_4_@PSS (325) > Fe_3_O_4_@Carb (151) > Fe_3_O_4_@Dex (63) > Fe_3_O_4_ (45) mg g^−1^. All samples showed no significant drug release (<7%) at pH 7.4, while at pH 5.0 the release rate and extent of desorption, although increased by 2–3 times, remained relatively low. The drug release percentage varies from 30 for Fe_3_O_4_@PSS to 13–16 for Fe_3_O_4_@Dex and Fe_3_O_4_@Carb, and up to about 6% for Fe_3_O_4_@PEI after a 30 min incubation at pH 5.0. Cytotoxicity analysis showed high biocompatibility of Fe_3_O_4_, Fe_3_O_4_@PSS, and Fe_3_O_4_@PEI. The survival rate of Neuro2A cells was above 80%. A preliminary assessment of the effect of unmodified and PSS-modified IONs on the rate of thrombus formation in the blood did not reveal noticeable changes. We believe that the obtained materials and results of the study can be taken into account when developing new drug delivery platforms.

## Figures and Tables

**Figure 1 ijms-24-04480-f001:**
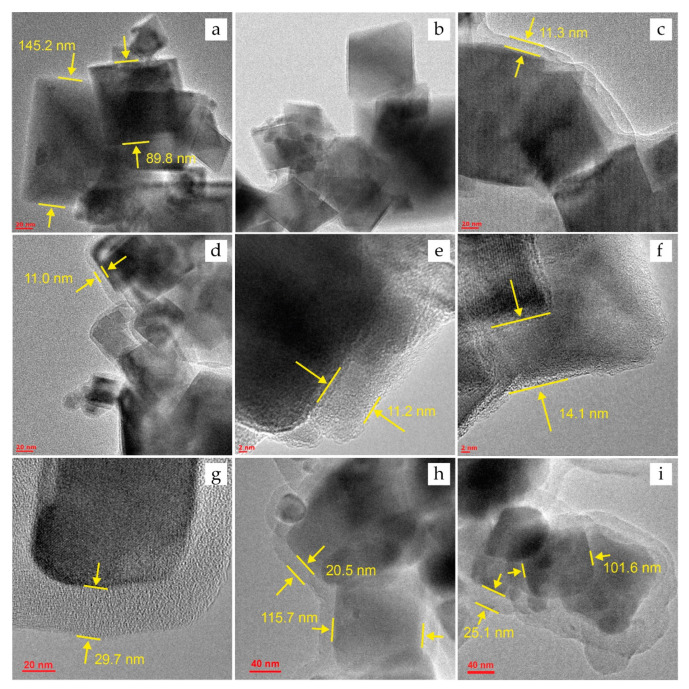
HRTEM images for the IONs before DOX sorption: (**a**)—Fe_3_O_4_, (**c**)—Fe_3_O_4_@PSS, (**e**)—Fe_3_O_4_@PEI, and (**g**,**h**)—Fe_3_O_4_@Carb; and after sorption of DOX: (**b**)—Fe_3_O_4_@DOX, (**d**)—Fe_3_O_4_@PSS@DOX, (**f**)—Fe_3_O_4_@PEI@DOX, and (**i**)—Fe_3_O_4_@Carb@DOX.

**Figure 2 ijms-24-04480-f002:**
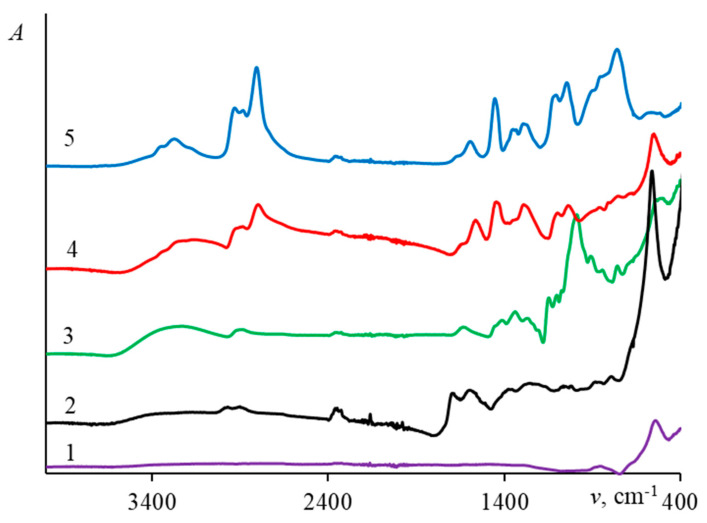
IR spectra of IONs covered by different surface modifiers: (1)—Fe_3_O_4_, (2)—Fe_3_O_4_@Carb, (3)—Fe_3_O_4_@Dex, (4)—Fe_3_O_4_@PEI, and (5)—Fe_3_O_4_@PSS.

**Figure 3 ijms-24-04480-f003:**
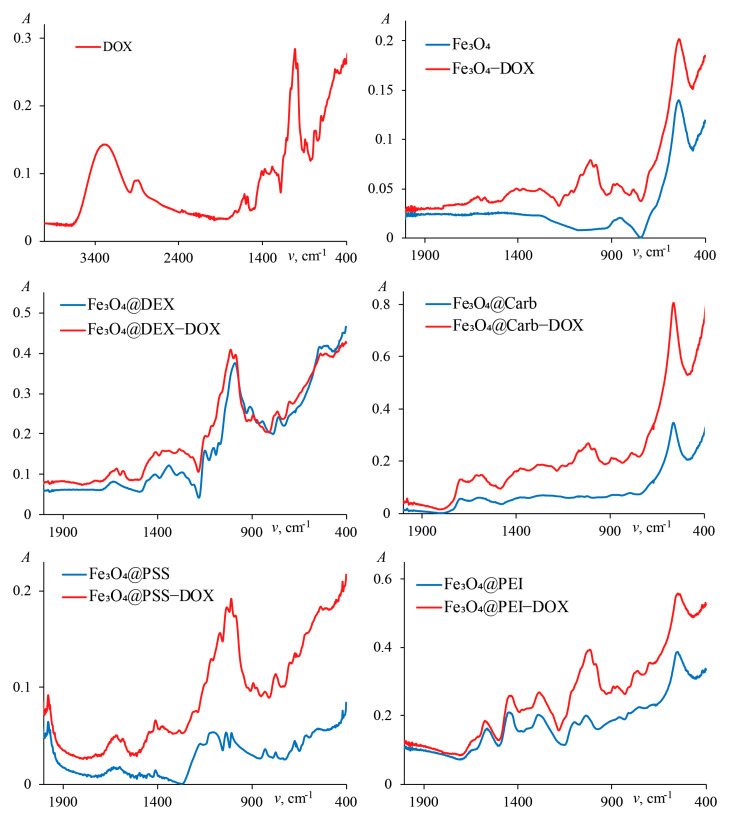
IR spectra of IONs covered by different surface modifiers with doxorubicin.

**Figure 4 ijms-24-04480-f004:**
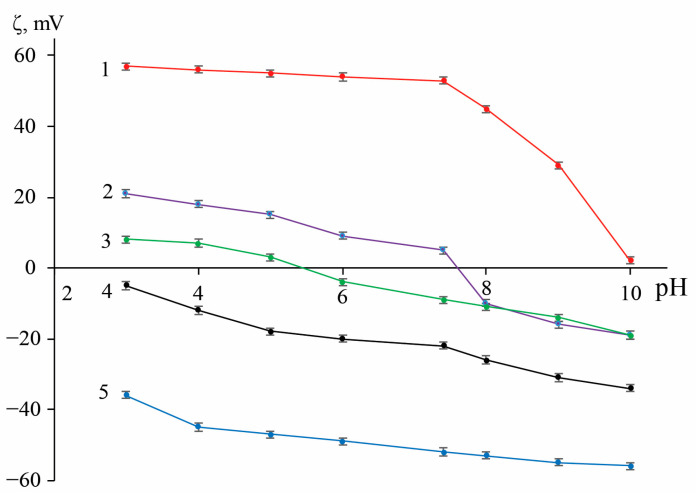
pH-responsive zeta-potential of original and functionalized IONs. Traces: (1)—Fe_3_O_4_@PSS, (2)—Fe_3_O_4_@Carb, (3)—Fe_3_O_4_@Dex, (4)—Fe_3_O_4_, and (5)—Fe_3_O_4_@PEI (*n* = 3, *p* = 0.95).

**Figure 5 ijms-24-04480-f005:**
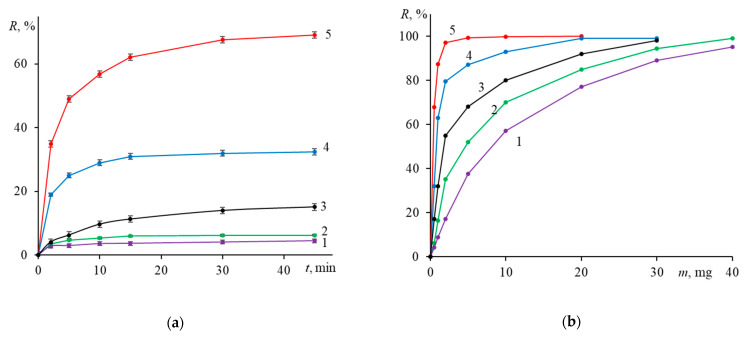
Time- and sorbent mass-dependent sorption of doxorubicin on various IONs. Conditions: (**a**) *m*_0_ = *m_S_* = 0.5 mg and (**b**) *m_S_* = 0.5 mg; time 45 min, pH 7.4, 25 °C. Traces: (1)—Fe_3_O_4_, (2)—Fe_3_O_4_@Dex, (3)—Fe_3_O_4_@Carb, (4)—Fe_3_O_4_@PSS, and (5)—Fe_3_O_4_@PEI. (*n* = 3, *p* = 0.95).

**Figure 6 ijms-24-04480-f006:**
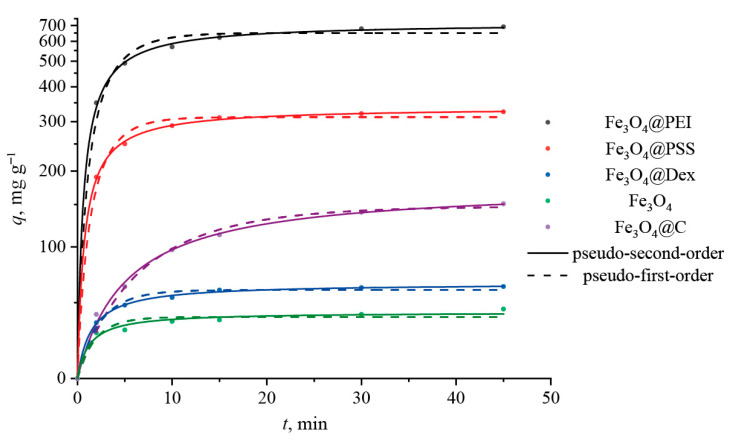
Adsorption kinetic curves of DOX onto different magnetite nanoparticles.

**Figure 7 ijms-24-04480-f007:**
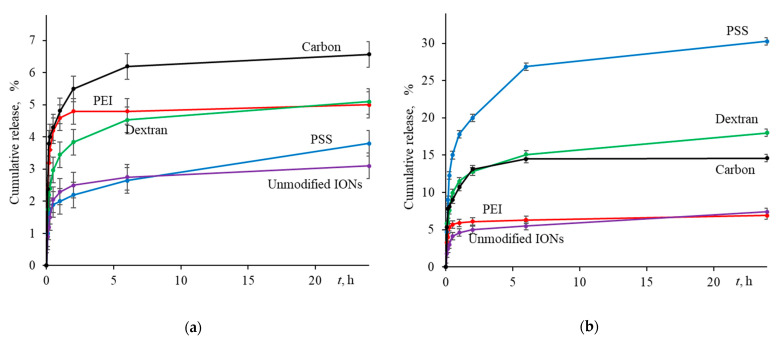
Payload release from drug-loaded nanosorbents in buffer solution at (**a**) pH 7.4 and (**b**) pH 5.0 as a function of time (*n* = 3, *p* = 0.95).

**Figure 8 ijms-24-04480-f008:**
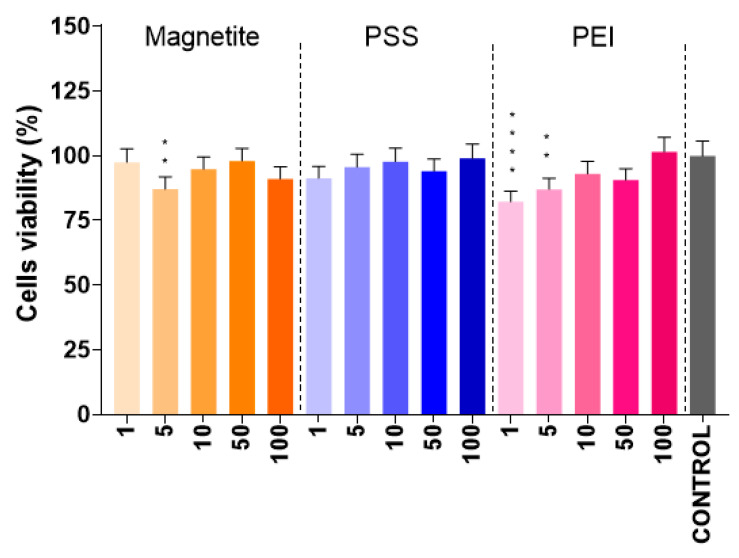
Results of the cell viability MTT test for N2A cells incubated with parent and modified IONs. The abscissa numbers represent concentration of nanoparticles (in µg·L^−1^); the *p* values (<0.01 and <0.0001) are marked, respectively, with, two, and four asterisks.

**Figure 9 ijms-24-04480-f009:**
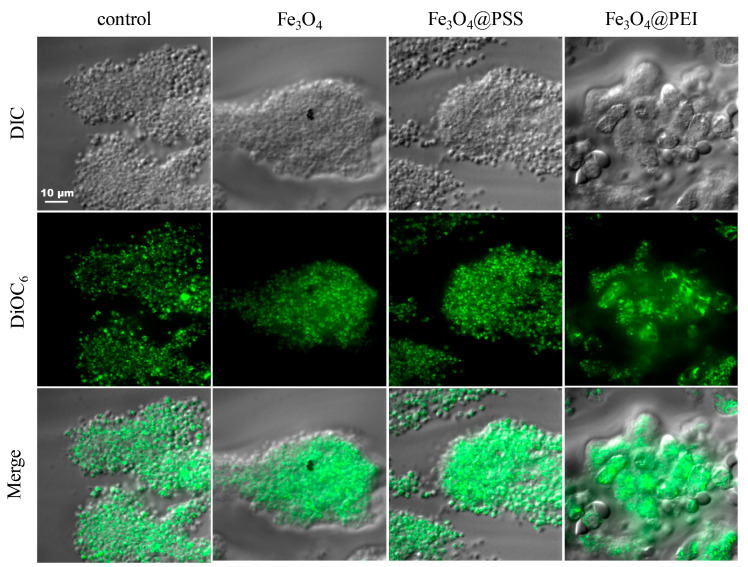
Micrographs of blood clots in the chamber in various shooting modes. DIC = differential interference contrast (optical microscope); DiOC_6_ = cell-permeant lipophilic dye that is selective for the mitochondria of live cells (fluorescence microscopy); Merge = summarized image (optical and fluorescent microscopy).

**Figure 10 ijms-24-04480-f010:**
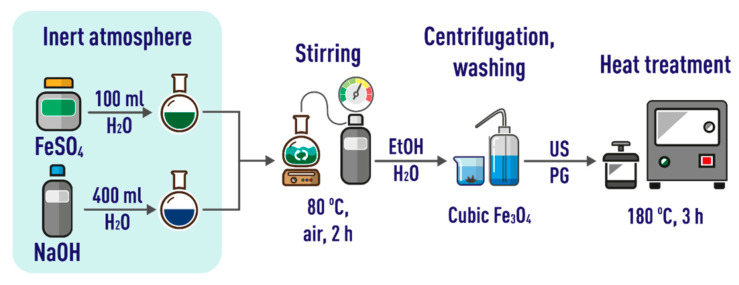
A workflow for obtaining nanoparticles of cubic magnetite. US = ultrasound; PG = propylene glycol.

**Figure 11 ijms-24-04480-f011:**
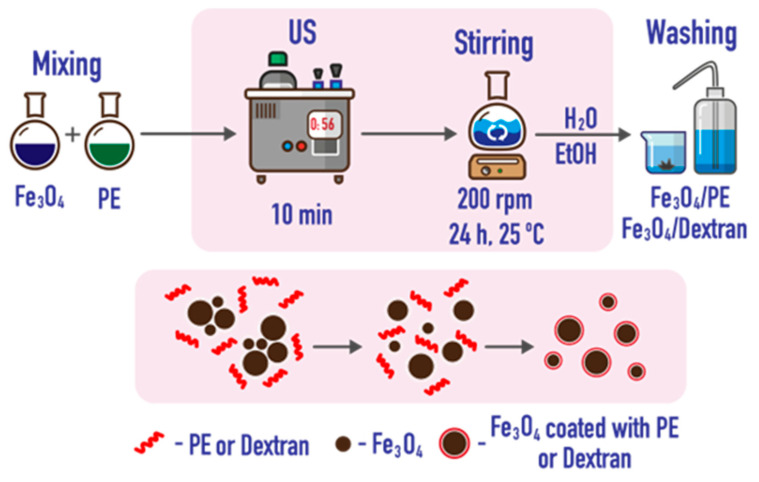
A scheme of modification of IONs with polyelectrolytes (PE) and dextran.

**Figure 12 ijms-24-04480-f012:**
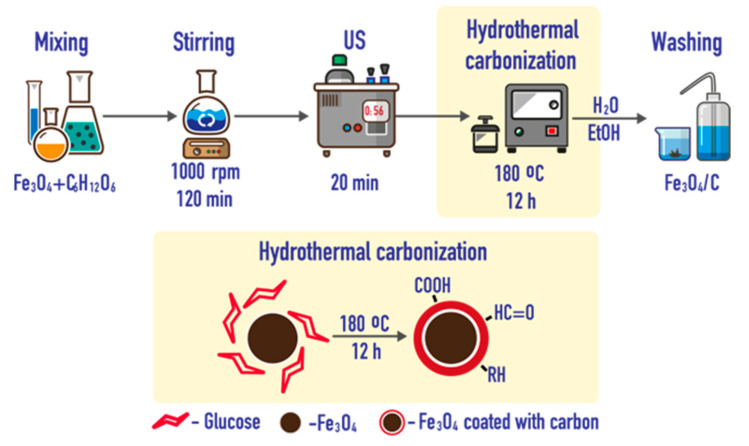
Synthesis of IONs coated with porous carbon.

**Figure 13 ijms-24-04480-f013:**
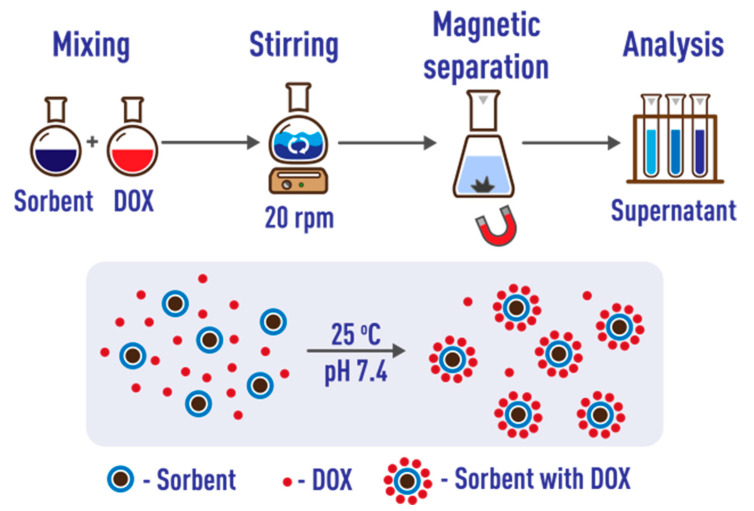
Exploring doxorubicin sorption on modified IONs. DOX = doxorubicin.

**Table 1 ijms-24-04480-t001:** IR band assignment for modified magnetite particles.

Surface Modifier	Wavenumber (cm^–1^)	Assignment
pure Fe_3_O_4_	575	Stretching vibrations of Fe–O bond [92]
PEI	3200–3400	Stretching vibrations of N–H bond [93]
	2920	Stretching vibrations of C–H groups of CH in PEI [94]
	2780	Stretching vibrations of C–H groups of CH_2_ in PEI [91]
	1560	Deformation vibrations of C–N bond
	1460	Deformation vibrations of CH_2_ group [95]
	1037,1090	Stretching vibrations of C–N bond [95,96]
Carb	3300–3450	Stretching vibrations of O–H bond
	2950, 2880	Stretching vibrations of C–H groups of CH in carbon [97]
	1700, 1580	Stretching vibrations of C=O bond [92,98]
	1604	Stretching vibrations of C=C bond [99]
	1000–1450	Stretching vibrations of C–O bond [98]
	875–750	Out-of-plane bending vibrations of aromatic CH groups [98]
Dex	3200	Stretching vibrations of O–H bond [100]
	2900	Stretching vibrations of C–H bond in -CH_2_ group [100]
	1636	Stretching vibrations of C=O [101]
	1344	Stretching vibrations of C–O [100]
	1105, 1075, and 1020–995,	Stretching vibrations of C–O–C ester group of dextran [100,102,103]
PSS	2930, 2800	Stretching vibrations of C–H bond in –CH_3_ and –CH_2_– groups [104]
	1590	Stretching vibration of a C–C bond in an aromatic ring
	1350	Vibrations of the O–S–O double bond in –SO_3_H group
	1200	Stretching vibrations of the O=S=O in –SO_3_H groups [105]
	1110	C–H bending vibrations of the aromatic ring
	1120, 1152, and 1034	S=O stretching vibrations of the sulfonic group [104,106]
	770	Bending vibrations of C–H

**Table 2 ijms-24-04480-t002:** Characteristic IR bands of doxorubicin and its conjugates with magnetite nanoparticles.

Wavenumber (cm^−1^)	Assignment
3282	Stretching vibrations of O–H bond
2925, 2885	Stretching vibrations of C–H bond in –CH_2_ group [107]
1725	Stretching vibrations of C=O [107]
1611, 1577, 1412	C=C stretching vibrations of the aromatic ring [107]
1105, 1067	Stretching vibrations of C–O–C ester group [107]
1008	Deformation vibrations of C=O [108]
693	Stretching vibrations C=C ring bend [107]

**Table 3 ijms-24-04480-t003:** Magnetic properties of IONs coated with various modifiers before and after sorption of DOX.

Sample	*Ms* (emu·g^−1^)		Δ (%)
	Before	After	
Fe_3_O_4_	77.7	77.6	<1
Fe_3_O_4_@PSS	71.1	69.3	3
Fe_3_O_4_@PEI	70.5	67.8	4
Fe_3_O_4_@Dex	70.2	70.0	<1
Fe_3_O_4_@Carb	60.8	58.4	4

**Table 4 ijms-24-04480-t004:** Colloidal stability, zeta-potential, and hydrodynamic size of aqueous suspensions with concentration 50 mg·mL^−1^ at 25 °C (*n* = 3, *p* = 0.95).

Sample	Average Settling Time (min)		ζ-Potential (mV)		Z-Average Size (nm) ^a^	
	pH 5.0	pH 7.4	pH 5.0	pH 7.4	pH 5.0	pH 7.4
Fe_3_O_4_	2.3 ± 0.1	2.0 ± 0.1 (2.1) ^b^	15 ± 1	5 ± 1	126 ± 4	151 ± 3
Fe_3_O_4_@PEI	8.5 ± 0.2	8.5 ± 0.2 (8.3)	55 ± 1	53 ± 1	95 ± 2	97 ± 2
Fe_3_O_4_@PSS	3.4 ± 0.1	10.3 ± 0.1 (10.1)	−47 ± 1	−52 ± 1	118 ± 5	118 ± 7
Fe_3_O_4_@Dex	<1	7.8 ± 0.2 (7.8)	3 ± 1	−9 ± 1	282 ± 9	130 ± 2
Fe_3_O_4_@Carb	~1	~1 (~1)	−18 ± 1	−22 ± 1	296 ± 8	290 ± 9

^a^ Measured immediately after evaluation of colloidal stability. ^b^ Given in parentheses is the average sedimentation time after the sorption of DOX.

**Table 5 ijms-24-04480-t005:** Kinetic parameters of the pseudo-first- and pseudo-second-order for the sorption of DOX at 25 °C.

Sample	*q_e_^exp^* (mg g^−1^)	Pseudo-First Order			Pseudo-Second Order		
		*q_e_* (mg g^−1^)	*k* _1_	*R^2^*	*q_e_* (mg g^−1^)	*k* _2_	*R* ^2^
Fe_3_O_4_/PEI	690 ± 10	649 ± 22	0.31 ± 0.05	0.972	719 ± 9	6.10 ± 0.04	0.998
Fe_3_O_4_/PSS	325 ± 10	311 ± 8	0.40 ± 0.05	0.982	337 ± 9	18.00 ± 0.01	0.999
Fe_3_O_4_/Carb	151 ± 8	147 ± 5	0.11 ± 0.01	0.986	178 ± 5	6.70 ± 0.08	0.996
Fe_3_O_4_/Dex	63 ± 7	60 ± 2	0.37 ± 0.05	0.979	66 ± 1	84.00 ± 0.06	0.998
Fe_3_O_4_	39 ± 8	39 ± 2	0.49 ± 0.15	0.913	42 ± 2	0.016 ± 0.005	0.965

## Data Availability

Not applicable.

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
