# Peer review of "Study on Doxorubicin Loading on Differently Functionalized Iron Oxide Nanoparticles: Implications for Controlled Drug-Delivery Application"

_ijms, 2023, doi:10.3390/ijms24054480_

Round 1

Reviewer 1 Report (New Reviewer)

The autors carried out a detailed study of the synthesis conditions, properties and applications of modified magnetic nanoparticles (carriers of doxorubicin), intended for use in cancer therapy. Conclusions about the possibility of using drugs have been confirmed by numerous modern instrumental methods. The research is carried out at a high scientific level and is well-presented. The paper can be published in its original form

Author Response

Reviewer 1: The autors carried out a detailed study of the synthesis conditions, properties and applications of modified magnetic nanoparticles (carriers of doxorubicin), intended for use in cancer therapy. Conclusions about the possibility of using drugs have been confirmed by numerous modern instrumental methods. The research is carried out at a high scientific level and is well-presented. The paper can be published in its original form. 

No comments, therefore, we have no reply. 

Reviewer 2 Report (Previous Reviewer 1)

I would like to express my appreciation for the modifications conducted by the authors. The manuscript quality has been improved significantly, and I strongly believe that it is suitable for publication. However, some minor comments before publications:

1- In the abstract, please revise the conclusion. I believe that the authors did not incorporate it appropriately.

2- Please remove the instrumentation for biological characterization from the section “2.5. Characterization of nanoparticles” and add it with detailed explanation in the section “2.9.2. Cell viability

Author Response

1- In the abstract, please revise the conclusion. I believe that the authors did not incorporate it appropriately.

We have changed the abstract and conclusions (see changes in red in the revised manuscript).

2- Please remove the instrumentation for biological characterization from the section “2.5. Characterization of nanoparticles” and add it with detailed explanation in the section “2.9.2. Cell viability

Done.

Reviewer 3 Report (Previous Reviewer 2)

The author has addressed all my comments. I will support this version for publication. 

Author Response

The author has addressed all my comments. I will support this version for publication. 

No comments, therefore no reply.

Reviewer 4 Report (New Reviewer)

The authors presented the paper "Study on doxorubicin loading on differently functionalized iron oxide nanoparticles: implications for controlled drug-delivery application"

Major comments

1) The reference list should be improved. Many more 2-3 years review papers should be cited in the Introduction section to show the progress in the area. I highly recommend not to use the references older 10 years for this section if it is possible (ref. 4, 9, 10, 14, 20, 39, 45, 46, 49, 50, 52, 55, 57, 59, etc.).

See the link on MDPI system search:

https://www.mdpi.com/search?sort=pubdate&page_count=50&q=magnetic+nanoparticles&year_from=2022&year_to=2023&featured=&subjects=&journals=nanomaterials%2Cpharmaceutics%2Cmagnetochemistry%2Ccancers%2Cencyclopedia&article_types=review-article%2CEntry&countries=

Related papers examples

https://www.mdpi.com/1999-4923/15/1/236

https://www.mdpi.com/2079-4991/12/20/3686

https://www.mdpi.com/2312-7481/9/1/12

https://www.mdpi.com/2079-4991/12/20/3567

https://www.mdpi.com/1999-4923/14/10/2093

https://www.mdpi.com/2079-4991/12/19/3323

https://www.mdpi.com/2673-8392/2/4/125

https://www.mdpi.com/2079-4991/12/21/3873

2) There are many experimental and review papers about doxorubicin and magnetic nanoparticles. See the link on MDPI system search:

https://www.mdpi.com/search?sort=pubdate&page_count=50&q=doxorubicin+magnetic+nanoparticles&year_from=2022&year_to=2023&featured=&subjects=chem-materials%2Cbio-life&journals=pharmaceutics%2Cnanomaterials%2Cpolymers%2Cmagnetochemistry%2Cijms&article_types=&countries=

I highly recommend to cite the recent works.

https://www.mdpi.com/2079-4991/12/20/3686

https://www.mdpi.com/2312-7481/8/5/54

https://www.mdpi.com/1999-4923/15/1/292

https://www.mdpi.com/2312-7481/8/10/114

https://www.mdpi.com/2079-4991/12/3/303

3) Section 2.6. Why you have used pH 7.4 for doxorubicin loading? Recent papers showed better loading in base media (pH 8-9). Why you have used such ION and doxorubicin concentrations? Have you tried to change doxorubicin concentration?

4) Why you have used phosphate buffer for loading and release? Phosphate ion may have specific properties and precipitation properties. Have you tried other salts? Please, mention phosphate buffer concentrations elsewhere.

5) Where supplementary material file with a number of figures?

6) In Figure 5, I see only nanoparticles agglomerates. Can you present figures with 1 μm magnification? If the nucleus of the nanoparticles is low, it doesn't mean that it will have good separated nanoparticles.

Can you present Malvern (DLS) size and Intensity Figures? 

7) Section 3.1.4 Can you explain somehow the changes in magnetization. It is important points for further studies. However, I don't see any explanations.

8) What is dav in Table 4. What method was used? DLS? If yes, you have much higher size data than you have presented in section 3.1.2.

9) Table 4. Is these setting times are good for your applications. Please, add some discussion in the text. It seems that is not very good. 

10) Have you studied the time-dependent stability of your nanoparticles at various pH, under storage in water, cell media, plasma? If no, please provide any stability data (size changes, etc.).

11) Section 3.3. Is it possible to provide a clear mechanism of doxorubicin sorption? What parameter is most important? Charge, hydrophobic chain, stacking, H-bonds? Why PEI and PSS coated nanoparticles have the highest capacity instead of others?

12) Section 3.5. I am sorry but I don't agree that PEI nanoparticles are the best. You have the same release at pH 7.4 and 5. It means that it is not pH-responsive nanoparticles. It means that your nanocomposite will release drug elsewhere (for example, in plasma) and may be cancer cells will not have any drug at all.

13) Branhed PEI with hgh positive charge usuall known as a toxic coating. We have used it in our prevous works in mouse model. There were some problems with a liver.  Please, mention cell types ang not ng but concentration in Figure 12. Moreover, you have lowest cell viability for magnetite using 10 ng,, For Fe3O4@PEI at 10 ng is the lowest toxicity. How you can explain it? Have you done the same experiments with doxorubicin-loaded nanoparticles?

did not statistically reduce cell survival - wrong statement. For PEI 0.1 ng ~ 75%, 10 ng -100%

Minor comments

1) You have high quality synthesis pictures. If Figures do not original, please cite the papers or multiplication software.

2) ref. 25 some data are missing.

3) Section 2.3 sonication power? Centrifugation rpm? Section 2.4 centrifugation rpm?

4) line 150 color from blue? not green?

Author Response

  1.  The reference list should be improved. Many more 2-3 years review papers should be cited in the Introduction section to show the progress in the area. I highly recommend not to use the references older 10 years for this section if it is possible (ref. 4, 9, 10, 14, 20, 39, 45, 46, 49, 50, 52, 55, 57, 59, etc.).

We have included new articles and shortened old references.

  1. There are many experimental and review papers about doxorubicin and magnetic nanoparticles.

We have updated the Introduction section.

  1. Section 2.6. Why you have used pH 7.4 for doxorubicin loading? Recent papers showed better loading in base media (pH 8-9). Why you have used such ION and doxorubicin concentrations? Have you tried to change doxorubicin concentration?

When choosing such pH conditions, we were guided by the literature data. Also, our choice of drug concentration was directed by the detectability of the method in use.

  1. Why you have used phosphate buffer for loading and release? Phosphate ion may have specific properties and precipitation properties. Have you tried other salts? Please, mention phosphate buffer concentrations elsewhere.

According to the literature, the phosphate buffer is commonly used in medicine. We did not use other buffer salts because it was not a part of research task. We did not found any data on the involvement of the phosphate ion in the precipitation of doxorubicin. As noted in the article, the precipitation of doxorubicin can be caused by both oxidation and dimerization of the drug [1]. However, we did not observe these phenomena.

  1. Yamada Y. Dimerization of Doxorubicin Causes Its Precipitation. ACS Omega. 2020, 5(51), p.33235-33241. doi: 10.1021/acsomega.0c04925.

  1. Where supplementary material file with a number of figures?

The file with Supplementary Materials is in the zip archive. To view it, you need to download the file and unzip it.

  1. In Figure 5, I see only nanoparticles agglomerates. Can you present figures with 1 μm magnification? If the nucleus of the nanoparticles is low, it doesn't mean that it will have good separated nanoparticles.

Can you present Malvern (DLS) size and Intensity Figures?

The Supplementary Materials contains low-resolution SEM photographs in μm magnification, which one can witness that the particles are not aggregated. We also added TEM images to the revised paper (Figure S3). From these TEM photographs, one may assume some aggregation but because the limitations of the method, we cannot conclude what the actual state of the particles in solution is. For a greater clarity, we have made additional changes in the text. Also, Figure 6S was added to the Supplementary Materials with the distribution of particle size and intensity distribution obtained by the DLS method.

  1. Section 3.1.4. Can you explain somehow the changes in magnetization. It is important points for further studies. However, I don't see any explanations.

Changes in magnetization are associated with the appearance of a non-magnetic coating that accords tendency described in literature. The coating shields the magnetic core from the action of an external magnetic field. As the coating thickness increases, the saturation magnetization decreases. Thus, the observed decrease in magnetization after the sorption of doxorubicin (table 3), in addition to the IR spectra, indicates successful sorption of the drug. This explanation is included in the revised text (p. 13).

  1. What is dav in Table 4. What method was used? DLS? If yes, you have much higher size data than you have presented in section 3.1.2.

The dav is the average particle diameter obtained by TEM and SEM (added to the text in section 3.1.2.).

  1. Table 4. Is these setting times are good for your applications. Please, add some discussion in the text. It seems that is not very good.

In Table 4, we present the results of the average settling time of particles under static conditions, when the solution is not stirred. Our goal was to evaluate the effect of the nature of the modifier on the deposition time of MNPs, and we obtained these data. For particles with an average size of more than 100 nm, this is an important parameter of colloidal stability. In our case, we observe a change in the average settling time depending on the type of the modifier and confirm the improvement in the stabilization of the polymer-coated dispersion. This explanation is added to the revised text (p. 14). 

  1. Have you studied the time-dependent stability of your nanoparticles at various pH, under storage in water, cell media, plasma? If no, please provide any stability data (size changes, etc.).

In this work, no such studies were carried out, leaving them room in our future research.  Our present focus was on the time-dependent stability of dispersions, including the stability of DOX-loaded particles, which refer to physiological pH values. The results showed that after the sorption of DOX, the stability of the dispersion was maintained.

  1. Section 3.3. Is it possible to provide a clear mechanism of doxorubicin sorption? What parameter is most important? Charge, hydrophobic chain, stacking, H-bonds? Why PEI and PSS coated nanoparticles have the highest capacity instead of others?

This is a very complex issue where other research and modeling methods are required. Based on our results, which refer to physiological pH values, we cannot say with certainty which mechanism is likely to dominate. However, we made an assumption included in the text (p. 15) and modified the text (p. 17). In a similar study, the authors believe that DOX interacts with PEI through a supramolecular mechanism involving a complex of specific and non-specific interactions. (Coluccini, C.; Ng, Y.M.; Reyes, Y.I.A. Functionalization of polyethyleneimine with hollow cyclotriveratrylene and its subsequent supramolecular interaction with doxorubicin. Molecules. 2020, 25(22), 5455;  https://doi.org/10.3390/molecules25225455) 

  1. Section 3.5. I am sorry but I don't agree that PEI nanoparticles are the best. You have the same release at pH 7.4 and 5. It means that it is not pH-responsive nanoparticles. It means that your nanocomposite will release drug elsewhere (for example, in plasma) and may be cancer cells will not have any drug at all.

We agree with this comment. The assignment of nanoparticles with PEI to the best is based on their high sorption capacity, which is 2–10 times higher than for other sorbents (Table 5). The absence of changes in release in the pH range of 5‒7.4 may be due to the fact that under these conditions 20‒50 percent of the PEI amino groups, which may be responsible for interacting with the DOX, remain unprotonated (see Ref. 128). In addition, the pH value in the local microenvironment of MNPs may differ from the pH in the bulk of the solution. From another point of view, a slow release of a drug may be more useful for medical applications where a slow release is required. To follow this point, we have corrected the text (p. 21) and conclusions so as not to mislead the reader.

13) Branhed PEI with hgh positive charge usuall known as a toxic coating. We have used it in our prevous works in mouse model. There were some problems with a liver.  Please, mention cell types ang not ng but concentration in Figure 12. Moreover, you have lowest cell viability for magnetite using 10 ng, For Fe3O4@PEI at 10 ng is the lowest toxicity. How you can explain it? Have you done the same experiments with doxorubicin-loaded nanoparticles?

did not statistically reduce cell survival - wrong statement. For PEI 0.1 ng ~ 75%, 10 ng -100%

We agree with comment. Recent studies in this area have shown that PEI has a number of disadvantages that may limit its use in vivo setting. Even in the platelet evaluation experiments performed, no satisfactory results were obtained. Further research in this area is needed. In this study, we were more interested in the issues of sorption and the behavior of doxorubicin at varying coatings. It is known that sorption goes better on surfaces with a negative charge. But in our studies, we found that the positively charged surface (or presence a lot of unprotonated amino groups PEI) plays a role. We have not yet conducted studies on cytotoxicity of MNPs with doxorubicin since this is a separate and large piece of new study.

Minor comments.

1) You have high quality synthesis pictures. If Figures do not original, please cite the papers or multiplication software.

All TEM and SEM micrographs in the article are original. The standard software supplied with the devices specified in section 2.5 was used.

2) ref. 25 some data are missing.

Corrected.

3) Section 2.3 sonication power? Centrifugation rpm? Section 2.4 centrifugation rpm?

Corrected.

4) line 150 color from blue? not green? 

Ferrous sulfate is green in dry form and blue in solution.

Round 2

Reviewer 4 Report (New Reviewer)

Thank you for the revised paper.

Author Response

Comments to reviewer

The reviewer did not provide specific comments. The decision was "minor revisions" in particular “minor spell check required”. We rewrote the “Conclusions” using the passive voice and the term "Thus". In section 3.2. we indicated intermolecular forces, which, according to our results and literature data, are responsible for the interaction of modifiers with magnetite.

These changes in the text are highlighted in blue.

This manuscript is a resubmission of an earlier submission. The following is a list of the peer review reports and author responses from that submission.

Round 1

Reviewer 1 Report

The authors have proposed a very important topic, which is the effect of different functionalization strategies of iron oxide nanoparticles (IONs) on the sorption of doxorubicin. I believe that the topic is timely and significant, and it adds to our knowledge when dealing with combining iron oxide nanoparticles in the drug delivery setting. However, I would like to respond to the authors with some major points that enhance both the experimental design, as well as addressing some claims that are not efficiently supported by data in this study.

1- Regarding the morphological aspects of IONs, the authors should provide the TEM images for each DOX-loaded IONs. Would sorption strategies could bring any changes in the cubic structures of IONs?

2- The IR spectra for free DOX and DOX-loaded IONs would represent important results to be included. It is significant to identify the characteristic peaks of DOX in each system. For instance, would the peaks of DOX change when it is incorporated in the porous shell structures of Fe3O4@C compared to other nanoparticles.

3- Here, your colloidal stability was measured in terms of particles sedimentation. What about the sedimentation and the zeta potential of the systems after the sorption of doxorubicin?

The long-term colloidal stability is a major concern. What about storage stability? Expectedly, it is an essential aspect of nanoparticles in a pharmaceutical setting. Would the functionalization strategies result in enhanced storage stability (changes in the particle sizes or zeta potentials)? Also, concerning the stability of DOX-loaded IONs, it should be measured.

4- In the release experiments, the authors used the same amount (by weight = 5 mg) of IONs, which may be unreliable. In fact, this can introduce bias, because the loading capacity of DOX is substantially different. It would be more reliable to use an equivalent amount of DOX, to investigate the effect of each sorbent in releasing its cargo. Also, if the nanoparticles were lyophilized, please state the methodology for lyophilization, the hydrodynamic particle size, and the zeta potential after reconstitution.

5- The sorption of DOX to Fe3O4@PEI is questionable. The desorption profile insignificantly changed compared to the un-modified Fe3O4. Especially, Fe3O4@PEI remains cationic at all pH ranges tested.

6- Iron oxide nanoparticles are potentially toxic. However, functionalization can alter their toxicity profile. In addition, the authors suggested that their systems can improve the efficacy of loaded DOX. Thereby, I suggest that the authors conduct an in vitro cytotoxicity study, as well as a cellular uptake study for each reported DOX-loaded IONs, in order to address their claims, regarding the choice of the system with the greatest medical potential.

Reviewer 2 Report

Comments:
The manuscript “Study on doxorubicin loading on differently functionalized iron oxide nanoparticles: implications for controlled drug-de-livery application” by Sergei N.Shtykov and colleagues study the adsorption and drug release kinetics of 5 different iron-oxide nanoparticles by loading with DOX. The author says the particles modified with PEI have better drug loading capacity, with PSS have greatest release at pH 5. The nanoparticles were well characterized. However, the reviewer believes that additional points of clarifications could potentially be addressed to further strengthen the manuscript.

1.       I am missing in this paper a more quantitative analysis of the drug loading experiments and that should definitely be done in a revision. This concerns the data shown in Fig. 7 which should be described in a quantitative way by an appropriate model for adsorption kinetics.

2.      Similarly, the data shown in Fig.8 should be described quantitatively by an appropriate adsorption isotherm.

3.      On Page 8, the author stated that the drug adsorption is due to the hydrogen bonds between non-protonated secondary amino groups of PEI and drug heteroatoms, please provide any experiment or citation to prove that this happened.

4.      In Fig.8 the drug release has been done in pH 5.0 saying the microenvironment inside cancer cells is slightly acidic. But if the particles uptake by cells, and it goes into endosome which is an acidic environment, then your drug released inside endosome and finally goes to lysosome without achieving any functions. How does author explain this?

5.      In Fig.8 I would recommend the author add another drug release study at pH 7.4 since this is more widely applied to biological applications.

6.      In Fig.8, the drug showed the burst release in first few minutes, I would recommend the author to reformate the x-axis in Fig.8 to show the clearer time points in the first few time points.

7.      In Fig.8, the data shows the burst release happened in 100 min, and there was nothing more release after that time point. This seems to me the drug will be hold inside the particles without release forever. I would recommend the author provide longer release time points, see if there is more drug release.

8.      The author says that the particles modified with PEI with highest loading capacity and modified with PSS have greatest release. But PEI and PSS are very toxic, it can also kill the normal cells, and thus, might have limited applications for in vivo study. How does the author explain this?

9.      I would also recommend the author to include the cell toxicity study in this manuscript.

Round 2

Reviewer 1 Report

Point 1: I do not agree with the authors. Investigating the morphological features of the IONs, especially after the sorption of DOX is significant. Also, the authors provided a reference where it was shown that the shell after modification was identifiable.

Please refer to reference: “2. Yoo H., Moon S-K., Hwang T., Kim Y.S., Kim J-H., Choi S-W., Kim J.H. Multifunctional Magnetic Nanoparticles Modified with Polyethylenimine and Folic Acid for Biomedical Theranostics. Langmuir. 2013, 29(20), 5962–5967. https://doi.org/10.1021/la3051302”.

Also, coating IONs in another reference, which was provided by the authors, has shown that modification has brought changes in the morphological features of the IONs.

Please refer to: “5. Zou C., Yao Y., Wei N., Gong Y., Fu W., Wang M., Jiang L., Liao X., Yin G., Huang Z., Chen X. Electromagnetic wave absorption properties of mesoporous Fe3O4/C nanocomposites. Composites Part B 2015. 77. 209-214. http://dx.doi.org/10.1016/j.compositesb.2015.03.030”.

Therefore, it would be imperative to record these changes, especially with at least drug adsorption. 

Point 3: With the assumption that the sedimentation stability after the sorption of DOX is the same for all systems. Please provide these data in table 3 in the manuscript. Also, indicate these results in the text.

Point 6: I strongly disagree with the authors, especially with the assumptions stated in the manuscript, which were not supported by any data about anticancer potential. In fact, in multiple examples provided by the authors, the assumptions of medical potential were supported by biological characterization involving Cell viability assay and cellular uptake. These experiments are considered very important to determine the medical potential of these systems. Please refer to the references (provided by the authors):

1- Yoo H, Moon SK, Hwang T, Kim YS, Kim JH, Choi SW, Kim JH. Multifunctional magnetic nanoparticles modified with polyethyleneimine and folic acid for biomedical theranostics. Langmuir. 2013 May 21;29(20):5962-7. doi: 10.1021/la3051302. Epub 2013 May 7. PMID: 23650947.

2- Javid A, Ahmadian S, Saboury AA, Kalantar SM, Rezaei-Zarchi S. Chitosan-coated superparamagnetic iron oxide nanoparticles for doxorubicin delivery: synthesis and anticancer effect against human ovarian cancer cells. Chem Biol Drug Des. 2013 Sep;82(3):296-306. DOI: 10.1111/cbdd.12145. Epub 2013 Jul 25. PMID: 23594157.

Reviewer 2 Report

The authors provided the revised manuscript with other languages there. This is a very unprofessional and should avoid next time. Overall, the author addressed a few points, but most of the points didn't address and says the editor didn't give enough time. I have a few comments:

1. No biological buffer release study performed. I'm still thinking pH 5.0 release didn't make sense for biological application. Your release study only performed in 1200 min, and the author said no time to do this experiment with biological buffer. 

2. Burst release was found, which is common, but drug release reached plateau within 1 h. Such rapid release time point has very limited biological applications. The author said the maximum release time is 24h in the literature, and says cells won't survive in longer treatment. Which literature you found says the cell won't survive for longer than 24 h? 

3. PEI and PSS is very toxic materials, the author didn't provide any toxicity experiment. 

Overall, due to lack of high and considerable importance and novelty, this work isn't suitable for publication in IJMS.